# Jietacin Derivative Inhibits TNF-α-Mediated Inflammatory Cytokines Production via Suppression of the NF-κB Pathway in Synovial Cells

**DOI:** 10.3390/ph16010005

**Published:** 2022-12-20

**Authors:** Kyoko Muneshige, Yuki Inahashi, Makoto Itakura, Masato Iwatsuki, Tomoyasu Hirose, Gen Inoue, Masashi Takaso, Toshiaki Sunazuka, Yoshihisa Ohashi, Etsuro Ohta, Kentaro Uchida

**Affiliations:** 1Department of Orthopedic Surgery, Kitasato University School of Medicine, 1-15-1 Minami-ku, Kitasato, Sagamihara City 252-0374, Japan; 2Graduate School of Infection Control Sciences, Kitasato University, 5-9-1 Minato-ku, Shirokane, Tokyo 108-8641, Japan; 3Ōmura Satoshi Memorial Institute, Kitasato University, 5-9-1 Minato-ku, Shirokane, Tokyo 108-8641, Japan; 4Department of Biochemistry, Kitasato University School of Medicine, 1-15-1 Minami-ku, Kitasato, Sagamihara City 252-0374, Japan; 5Department of Immunology II, Kitasato University School of Allied Health Sciences, 1-15-1 Minami-ku, Kitasato, Sagamihara City 252-0375, Japan; 6Shonan University of Medical Sciences Research Institute, Nishikubo 500, Chigasaki 253-0083, Japan

**Keywords:** jietacin derivative, NF-κB, osteoarthritis, synovial inflammation

## Abstract

Synovial inflammation plays a central role in joint destruction and pain in osteoarthritis (OA). The NF-κB pathway plays an important role in the inflammatory process and is activated in OA. A previous study reported that a jietacin derivative (JD), (*Z*)-2-(8-oxodec-9-yn-1-yl)-1-vinyldiazene 1-oxide, suppressed the nuclear translocation of NF-κB in a range of cancer cell lines. However, the effect of JD in synovial cells and the exact mechanism of JD as an NF-κB inhibitor remain to be determined. We investigated the effect of JD on TNF-α-induced inflammatory reaction in a synovial cell line, SW982 and human primary synovial fibroblasts (hPSFs). Additionally, we examined phosphorylated levels of p65 and p38 and expression of importin α3 and β1 using Western blotting. RNA-Seq analysis revealed that JD suppressed TNF-α-induced differential expression: among 204 genes significantly differentially expressed between vehicle and TNF-α-stimulated SW982 (183 upregulated and 21 downregulated) (FC ≥ 2, Q < 0.05), expression of 130 upregulated genes, including inflammatory cytokines (*IL1A*, *IL1B*, *IL6*, *IL8*) and chemokines (*CCL2*, *CCL3*, *CCL5*, *CCL20*, *CXCL9*, 10, 11), was decreased by JD treatment and that of 14 downregulated genes was increased. KEGG pathway analysis showed that DEGs were increased in the cytokine–cytokine receptor interaction, TNF signaling pathway, NF-κB signaling pathway, and rheumatoid arthritis. JD inhibited *IL1B*, *IL6* and *IL8* mRNA expression and IL-6 and IL-8 protein production in both SW982 and hPSFs. JD also suppressed p65 phosphorylation in both SW982 and hPSFs. In contrast, JD did not alter p38 phosphorylation. JD may inhibit TNF-α-mediated inflammatory cytokine production via suppression of p65 phosphorylation in both SW982 and hPSFs. Our results suggest that JD may have therapeutic potential for OA due to its anti-inflammatory action through selective suppression of the NF-κB pathway on synovial cells.

## 1. Introduction

Osteoarthritis (OA) is the most common type of arthritis, and an important cause of disability that degrades quality of life and causes substantial economic loss [1]. A number of guidelines for management of osteoarthritis (OA) pain in patients presenting with severe pain and musculoskeletal pain universally recommend oral nonsteroidal anti-inflammatory drugs (NSAIDs) [2,3,4,5]. However, NSAIDs are restricted to pain management rather than prevention or cure, leaving surgery as typically the last resort for knee OA. Therefore, the development of disease-modifying drugs that confer both structural and symptomatic benefits is needed for OA treatment.

Synovial inflammation plays a pivotal role in joint destruction and pain in osteoarthritis. The inducible transcription factor NF-κB has a central role in immune and inflammatory responses, and in cellular differentiation. In particular, expression of NF-κB is upregulated in synovial tissues in early and late OA [6,7]. Inflammatory cytokines, including IL-6 and IL-8 derived from the OA synovium, are well-known causes of catabolic gene induction operating via mechanisms which involve NF-κB activation [8,9]. In addition, these cytokines contribute to osteoarthritic pain [10,11]. Moreover, NF-κB regulates chemokine ligands to recruit immune cells during synovial inflammation [12,13]. Therefore, the development of NF-κB inhibitors may provide a potential therapeutic agent for OA.

Previous studies reported the potential of natural products in the treatment of OA via the inhibition of inflammatory processes [14,15]. Jietacins, a class of azoxy antibiotics, were originally isolated from culture broth of *Streptomyces* sp. KP-197 [16,17]. We previously reported that a jietacin derivative (JD), (Z)-2-(8-oxodec-9-yn-1-yl)-1-vinyldiazene 1-oxide (Appendix A), suppressed the nuclear translocation of NF-κB in a range of cancer cell lines having a strong constitutive NF-κB activation [18]. However, the effect of JD on inflammatory reaction in synovial cells and the exact mechanism of JD as an NF-κB inhibitor remain to determined.

Here, we examined the effect of JD on inflammatory cytokine production and phosphorylation of NF-κB in synovial cells.

## 2. Results and Discussion

### 2.1. Effect of JD on TNF-α-Stimulated Inflammatory Reaction in Synovial Cells

To evaluate the effect of JD on the TNF-α-stimulated inflammatory response, we performed RNA-Seq analysis using synovial cell line SW982. Differential expression genes (DEGs) analysis (FC ≥ 2, Q < 0.01) revealed that 204 genes (183 upregulated and 21 downregulated) were significantly differentially expressed between vehicle and TNF-α-stimulated SW982 (Figure 1A, Appendix A). Of these DEGs, 130 upregulated genes, including inflammatory cytokines (*IL1A*, *IL1B*, *IL6*, *IL8)* and chemokines (*CCL2*, *CCL3*, *CCL5*, *CCL20*, *CXCL9*, *10*, *11*), were reduced by JD treatment, and 14 downregulated genes were increased by JD treatment (Figure 1A, Appendix A). Pathway analysis showed that DEGs were increased in the cytokine–cytokine receptor interaction (KEGG pathway ID, 04060), TNF signaling pathway (KEGG pathway ID, 04668), NF-κB signaling pathway (KEGG pathway ID, 04060), and rheumatoid arthritis (KEGG pathway ID, 05323) (Table 1, Figure 1B).

To validate the results of the RNA-Seq analysis and to determine the effect of JD concentrations, we examined the effect of various JD concentrations on inflammatory cytokine production using qPCR and ELISA in SW982 (Figure 2A–D). Stimulation of SW982 with hrTNF-α significantly increased *IL1B* mRNA expression and supernatant IL-1β protein levels (*IL1B*, *p* < 0.001; IL-1β *p* < 0.001; Figure 2A,B). Exposure to 1.25 and 2.5 μg/mL JD inhibited this increase (1.25 μg/mL, *IL1B*, *p* < 0.001. IL-1β *p* = 0.005; 2.5 μg/mL, *IL1B*, *p* < 0.001, IL-1β, *p* < 0.001; Figure 2A,B). Stimulation of SW982 with hrTNF-α significantly also increased *IL6* mRNA expression and supernatant IL-6 protein levels (*IL6*, *p* < 0.001; IL-6 *p* = 0.003; Figure 2C,D). Exposure to 1.25 and 2.5 μg/mL JD inhibited this increase (1.25 μg/mL, *IL6*, *p =* 0.002. IL-6 *p* = 0.011; 2.5 μg/mL, *IL6*, *p* < 0.001, IL-6, *p* < 0.001; Figure 2C,D). Further, stimulation of SW982 with hrTNF-α also significantly increased *IL8* mRNA expression and supernatant IL-8 protein levels (*IL8*, *p* = 0.001; IL-8 *p* = 0.004; Figure 2E,F), and exposure to 2.5 μg/mL JD inhibited this increase (*IL8*, *p* < 0.001. IL-8, *p* < 0.001. Figure 2E,F).

Based on the results in SW982, we next investigated the anti-inflammatory effect on human primary synovial fibroblasts (hPSFs). TNF-α stimulated *IL1B* (*p* < 0.011), *IL6* (*p* < 0.011), and *IL8* mRNA (*p* < 0.001) expression was also significantly reduced by JD treatment (*IL1B, p* < 0.001; *IL6, p* < 0.001; *IL8*, *p* < 0.001) in hPSFs (Figure 3A–C). However, IL-1β was below the detection limit of the ELISA (<2 pg/mL) (Figure 3D), consistent with a previous study [19]. IL-6 and IL-8 production increased following TNF-α treatment (IL-6, *p* < 0.001; IL-8, *p* < 0.001; Figure 3E,F) and reduced these protein levels in supernatant in the presence of JD (IL-6, *p* < 0.001; IL-8, *p* < 0.001; Figure 3E,F).

Activated NF-κB increases the expression of inflammation-related cytokines and chemokines in synovial fibroblasts [12,13,20], and these expressions are associated with OA pathology [10,11,20,21,22,23,24,25]. CCL2 and CCL3 are elevated in peripheral blood in OA patients [23], and CXCL9 and CXCL11 are increased in synovial fluids (SF) and serum of OA patients [24]. CCL20 concentrations in SF correlate with disease severity in knee OA [25]. IL-1β concentration in SF is dependent on OA grade [26]. Serum IL-6 levels are increased in patients with OA, and are associated with radiological OA grade [20,27]. In addition, level of IL-8 in SF is associated with OA severity [21]. In addition, several clinical studies in OA patients have demonstrated a correlation between pain score and concentrations of IL-6 and IL-8 in SF [10,11]. Our results suggest that JD may have therapeutic potential for OA due to its anti-inflammatory action.

### 2.2. Effect of JD on NF-κB Pathway

KEGG pathway analysis indicated that JD regulated the NF-κB pathway. In addition, given that the JD suppressed TNF-α-mediated IL-1β, IL-6, and IL-8, we subsequently examined the effect of JD on the NF-κB pathway. TNF-α induced the phosphorylation of p65 (*p* < 0.0001), which was suppressed in the presence of 2.5 μg/mL (*p* < 0.001) and 1.25 μg/mL (*p* = 0.002) JD (Figure 4A,B). There was no difference in p65 expression among the groups (Figure 4A,C). Consistent with the SW982 results, TNF-α induced the phosphorylation of p65 (*p* < 0.0001), which was suppressed in the presence of 2.5 μg/mL JD (*p* < 0.001) in hPSFs (Figure 5A,B).

The mammalian NF-κB subfamily comprises five proteins, namely, p50 (NF-κB), p52 (NF-κB2), p65 (RelA), RelB, and c-Rel [22]. NF-κB is retained in the cytosol in an inactive state under normal conditions, bound to the inhibitory protein IκB [28,29]. Any or all of proinflammatory cytokines, matrix degradation enzymes, and excessive mechanical stress can induce a cascade of reactions which lead to the phosphorylation of IκB and resulting proteasome-system-mediated degradation via the ubiquitin proteasome system. Further, IκB degradation then leads to the release of active NF-κB, which is translocated to the nucleus where it induces gene transcription [28]. We previously reported that JD inhibits this translocation to the nuclei, and that its inhibitory effect is lost in cells having mutant p65 protein [18]. Phosphorylated p65 at this position plays the pivotal role of terminating the transcriptional activity of NF-κB in nuclei [7]. JD, therefore, prevents the transcriptional activity of NF-κB via the inhibition of p65 phosphorylation.

Nucleocytoplasmic shuttling of NF-κB is a tightly controlled system which is mediated by specific nuclear import/export process. Two types of importins have been identified: importin α, with seven members in mammals [30,31,32,33,34,35], and importin β, with >20 members in mammals [36,37,38]. A recent study reported the translocation of p50/p65 into the nucleus via adapter importin α3 and receptor importin β [39]. Another study reported that import of p65 relies mainly on KPNB1 (importin β1) [40]. No significant difference was observed in importin α3 among the groups (Figure 4A,D), similar to our previous study [16]. In contrast, importin β1 expression in SW982 was decreased in the presence of 2.5 μg/mL (*p* < 0.001) and 1.25 μg/mL (*p* = 0.002) JD (Figure 4A,E). However, importin α3 and β1 were not altered in the presence of 2.5 μg/mL JD in hPSFs (Figure 5A,D,E). A previous study reported that TNF-α-mediated signaling partly differed between SW982 and human primary synovial fibroblasts (hPSFs) [19]. Effect of JD on importin β1 differed between SW982 and hPSFs, and decreased importin β1 in SW982 may not have contributed to the suppression of inflammatory production.

TNF-α activates a number of intracellular signaling pathways, among which are the NF-κB and p38 pathways [41,42,43]. A previous study reported that p38 plays a role in TNF-α-induced production of inflammatory cytokines independently of the NF-κB pathway in synovial fibroblasts [43]. TNF-α induced phosphorylation of p38 in both the presence and absence of JD in SW982 (*p* < 0.001; Figure 6A,B). However, no change was seen in phosphorylation of p38 expression in the presence of JD (Figure 6A,C). Similarly, TNF-α-stimulated phosphorylation of p38 (Figure 7A,B, *p* < 0.001) was not suppressed by JD in hPSFs (Figure 7A,C). JD suppresses production of inflammatory cytokines by selectively inhibiting p65 phosphorylation.

## 3. Materials and Methods

### 3.1. Synthesis of JD

A jietacin derivative (JD), (*Z*)-2-(8-oxodec-9-yn-1-yl)-1-vinyldiazene 1-oxide, was synthesized using a method we previously reported [18,19].

### 3.2. Cell Culture

Human synovial cell line SW982 was purchased from American Type Culture Collection (Rockville, MD, USA). hPSFs were purchased from Sigma Aldrich (Sigma-Aldrich, St. Louis, MO, USA). Cells were cultured in Dulbecco’s Modified Eagle Medium with supplementation with 10% fetal bovine serum, 250 ng/mL amphotericin B, 100 ng/mL streptomycin, and 100 U/mL penicillin at 37 °C in 5% CO_2_.

### 3.3. RNA-Seq

RNA-Seq assay was used to screen the effect of JD on TNF-α stimulated SW982 cells. SW982 cells were seeded in 2 mL at a concentration of 1 × 105 cells/mL in a 6-well plate and incubated for 72 h. The cells were then treated with either control medium (vehicle) or TNF-α in the presence of 2.5 μg/mL JD for 6 h.

The total RNA was obtained using the Trizol protocol (Invitrogen, Carlsbad, CA, USA) and spin column (Direct-zol MicroPrep kit, Zymo Research, Orange, CA, USA). RNA quantity was determined by spectrophotometer (Denovix, DX, USA) and quality was assessed on an Agilent 2100 BioAnalyzer (Agilent) with an RNA 6000 Nano Chip. RNA-Seq was conducted on extracted RNA. RNA sequencing was performed using an MGI DNBSEQ-G400 sequencer (BGI, Shenzhen, China). Two replicate samples were taken from the vehicle, TNF-α, and TNF-α + JD groups for RNA-Seq.

### 3.4. Quantitative PCR (qPCR)

To validate the results of the RNA-Seq analysis and to investigate the effect of JD concentrations, qPCR was used to evaluate expression levels of *IL1B*, *IL6* and *IL8*. SW982 cells and hPSFs were seeded in 2 mL at a concentration of 1 × 10^5^ cells/mL in a 6-well plate and incubated for 72 h. SW982 cells were then treated with either control medium (vehicle) or TNF-α in the presence of various concentration of JD (0, 0.625, 1.25, and 2.5 μg/mL) for 6 h (*n* = 8). hPSFs were then treated with either control medium (vehicle) or TNF-α in the presence of 2.5 μg/mL JD for 6 h (*n* = 8). Following RNA extraction as described above, first-strand cDNA was synthesized from purified total RNA using the SuperScript^®^ III First-Strand Synthesis System (Invitrogen). qPCR was conducted using the SYBR green method using the CFX connect real-time PCR detection system (Bio-Rad, Hercules, CA, USA). Table 2 lists the primer sequences used in this study. We evaluated GAPDH as a housekeeping gene. Gene expression (Gene/GAPDH) was measured by the delta–delta CT method, and relative expression was determined when the average level of gene expression (Gene/GAPDH) in vehicle was 1.

### 3.5. Enzyme-Linked Immunosorbent Assay (ELISA)

SW982 cells and hPSFs were seeded at a concentration of 1 × 10^5^ cells/mL in 100 μL in 96-well plates followed by incubation for 72 h. SW982 cells were then treated with TNF-α in the presence of various concentrations of JD (0, 0.625, 1.25 and 2.5 μg/mL) for 24 h (*n* = 8). hPSFs were then treated with TNF-α in the presence of 2.5 μg/mL JD for 24 h (*n* = 8). The supernatants were collected and IL-1β, IL-6 and IL-8 concentrations in the supernatants were measured using commercial ELISA kits (BioLegend, San Diego, CA, USA) according to the manufacturer’s protocol.

### 3.6. Western Blot

The SW982 cells and hPSFs were seeded at a concentration of 1 × 10^5^ cells/mL in 2 mL in a 6-well plate and incubated for 72 h. The SW982 cells were treated with TNF-α in the presence of several concentrations of JD (0, 0.625, 1.25 and 2.5 μg/mL) for 15 min (*n* = 3). The hPSFs were treated with TNF-α in the presence of JD (0 and 2.5 μg/mL) for 15 min (*n* = 3). Following treatment, total protein was extracted using sodium dodecyl sulfate (SDS) sample buffer, and the homogenates were immediately heated for 10 min at 95 °C. Protein concentrations were measured using a bicinchoninic acid (BCA) assay kit (Thermo Fisher Scientific, Inc., Waltham, MA, USA). The homogenates (3 μg/lane) were subject to SDS-polyacrylamide gel electrophoresis. Proteins separated in the gel were transferred electrophoretically onto a polyvinylidene difluoride membrane in blotting buffer, which was then treated with 10% skim milk in TBST at 25 °C for 30 min to cancel nonspecific reactions. It was then incubated with anti-p65 mouse monoclonal antibody (1:10,000; cat no. #6956, Cell Signaling Technology, Boston, MA, USA), anti-phospho-p65 (Ser536) rabbit monoclonal antibody (1:10,000; cat no. #76778, Cell Signaling Technology), anti-importin α3 (affinity-purified rabbit polyclonal importin α3 antibody was raised against a synthetic peptide (C)GFNSSTNVPTEGFQF corresponding to the mouse importin α3 C-terminal region), anti-importin β1 rabbit monoclonal antibody (1:1000; cat no. #51186), anti-p38 MAPK (1:1000; cat no. #9212, Cell Signaling Technology), anti-phospho-p38 MAPK (Thr180/Tyr182) (1:1000; cat no. 9211, Cell Signaling Technology), or anti-GAPDH antibody (1:5000; FUJIFILM Wako Pure Chemical Co., Osaka, Japan) at 25 °C for 60 min. Following further incubation with HRP-conjugated anti-mouse antibody (catalog number. 115-035-146, FUJIFILM Wako Pure Chemical Co.; West Grove, PA, USA) or goat anti-rabbit antibody (catalog number. 115-035-003, FUJIFILM Wako Pure Chemical Co.) for 60 min at 25 °C, the membrane was again washed for a final time, and protein bands were visualized with enhanced chemiluminescence (catalog number 07880, Chemi-Lumi One L; Nacalai Tesque, Kyoto, Japan) via luminescent image analysis using a CCD imager (LAS-4000mini; Fuji Photo Film Co., Tokyo, Japan). Relative expression (proteins/GAPDH) was analyzed using ImageJ software.

### 3.7. Statistical Analysis

Differences among vehicle-, TNF-α-, and TNF-α and JD-treated cells were compared using Tukey’s multiple comparisons test on SPSS (version 25.0, IBM, Armonk, NY, USA). All statistical analyses were two-sided. Statistical significance was indicated by a *p* value < 0.05.

## 4. Conclusions

JD inhibited TNF-α-mediated inflammatory reaction via the suppression of p65 phosphorylation in the synovial cell line SW982 and hPSFs. Our results suggest that JD may have therapeutic potential for OA due to its anti-inflammatory action through selective suppression of the NF-κB pathway on synovial cells. Further investigation using OA animal models may reveal therapeutic potential for OA.

## Figures and Tables

**Figure 1 pharmaceuticals-16-00005-f001:**
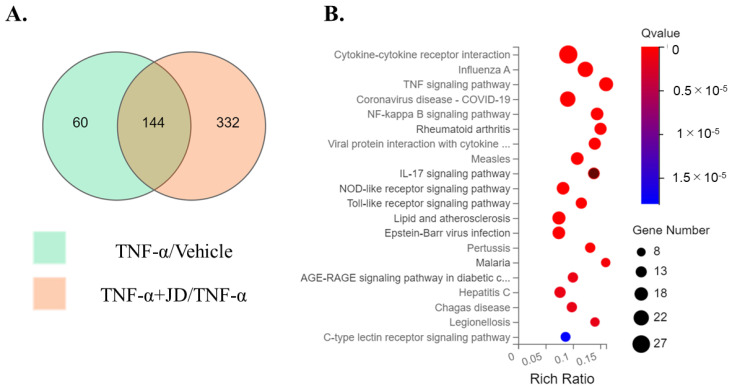
RNA-Seq analyses of DEGs in vehicle, TNF-α, and TNF-α + JD groups. (**A**) Venn diagram analysis of DEGs. The region of overlap among the two circles represents the common DEGs between TNF-α/vehicle and TNF-α + jietacin derivative (JD)/TNF-α datasets. DEGs, differentially expressed genes. (**B**) Bubble diagram of 20 enrichment KEGG pathways of DEGs.

**Figure 2 pharmaceuticals-16-00005-f002:**
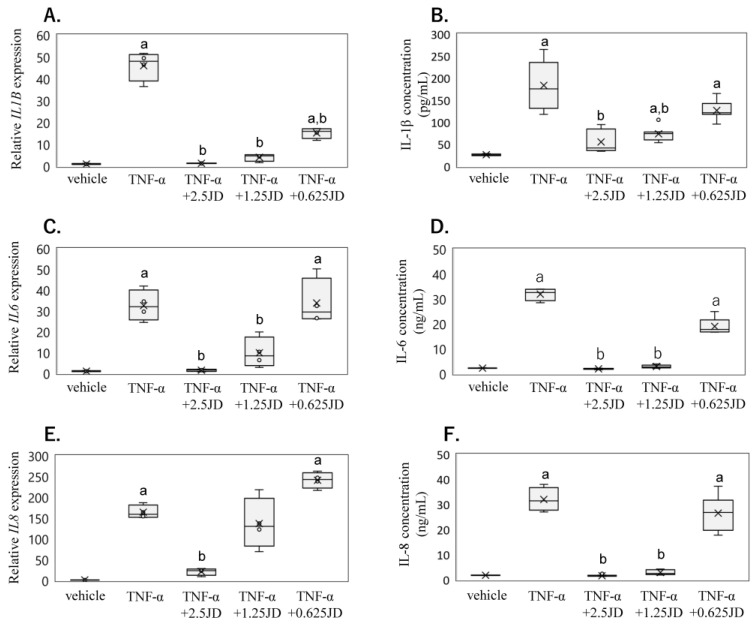
Effect of jietacin derivative on inflammatory cytokine expression and production in SW982. *IL1B* mRNA by qPCR (**A**) and IL-1β protein concentration by ELISA (**B**). *IL6* mRNA by qPCR (**C**) and IL-6 protein concentration by ELISA (**D**). *IL8* mRNA by qPCR (**E**) and IL-8 protein concentration by ELISA (**F**). Synovial cell line SW982 stimulated with DMEM (vehicle), TNF-α, or TNF-α + jietacin derivative. ^a^
*p* < 0.05 compared to vehicle, ^b^
*p* < 0.05 compared to TNF-α.

**Figure 3 pharmaceuticals-16-00005-f003:**
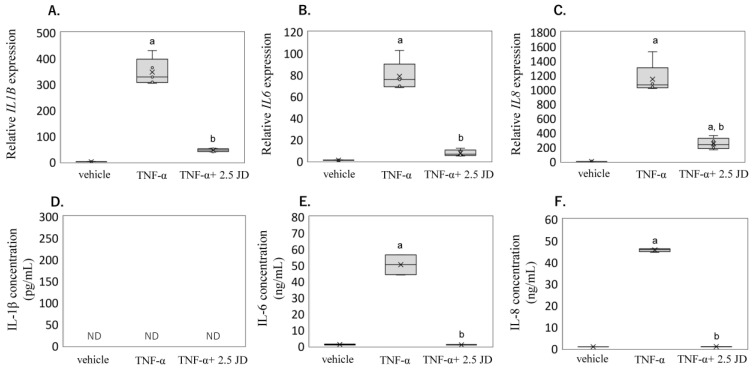
Effect of jietacin derivative on inflammatory cytokine expression and production in human primary synovial fibroblasts. *IL1B* (**A**), *IL6* (**B**), *IL8* (**C**) mRNA by qPCR, and IL-1β (**D**), IL-6 (**E**), IL-8 (**F**) protein concentration by ELISA. Human primary synovial fibroblasts stimulated with DMEM (vehicle), TNF-α, or TNF-α + jietacin derivative. ^a^
*p* < 0.05 compared to vehicle, ^b^
*p* < 0.05 compared to TNF-α.

**Figure 4 pharmaceuticals-16-00005-f004:**
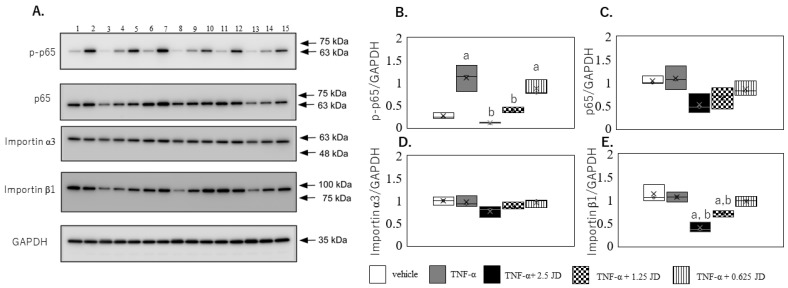
Effect of jietacin derivative on the NF-κB pathway in SW982. Western blotting for p65, phosphorylated p65 (p-p65), importin α3, importin β1 and GAPDH (**A**). Densitometry of western blot protein bands for p-p65 (**B**), p65 (**C**), importin α3 (**D**), and importin β1 (**E**) were normalized to the expression of GAPDH (*n* = 3). Vehicle, Lane 1,6,11; TNF-α, Lane 2,7,12; TNF-α+ 2.5 JD, Lane 3,8,13; TNF-α+ 1.25 JD, Lane 4,9,14; TNF-α+ 0.625 JD, Lane 5,10,15. ^a^
*p* < 0.05 in comparison with vehicle, ^b^
*p* < 0.05 compared to TNF-α.

**Figure 5 pharmaceuticals-16-00005-f005:**
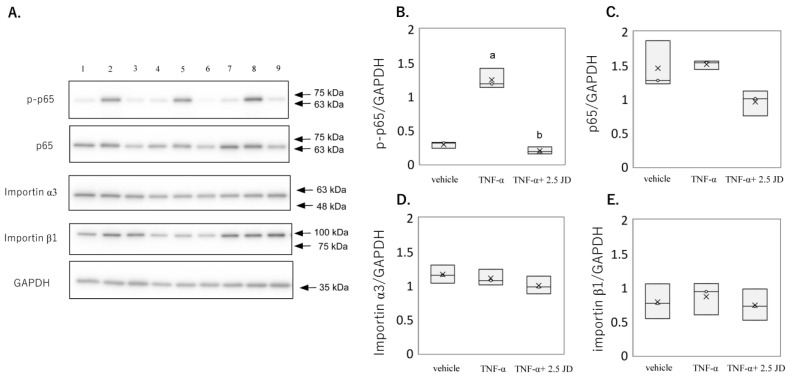
Effect of jietacin derivative on the NF-κB pathway in human primary synovial fibroblasts. Western blotting for p65, phosphorylated p65 (p-p65), importin α3, importin β1, and GAPDH (**A**). Densitometry of Western blot protein bands for p-p65 (**B**), p65 (**C**), importin α3 (**D**), and importin β1 (**E**) were normalized to the expression of GAPDH (*n*  = 3). Vehicle, Lane 1,4,7; TNF-α, Lane 2,5,8; TNF-α+ 2.5 JD Lane 3,6,9. ^a^
*p* < 0.05 compared with vehicle, ^b^
*p* < 0.05 compared to TNF-α.

**Figure 6 pharmaceuticals-16-00005-f006:**
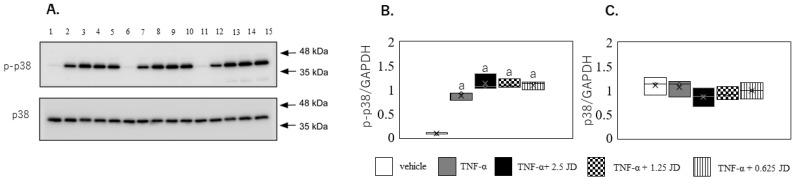
Effect of jietacin derivative on the p38 pathway in SW982. Western blotting for p38, phosphorylated p38 (p-p38), and GAPDH (**A**). Densitometry of Western blot protein bands for p-p38 (**B**) and p38 (**C**) were normalized to the expression of GAPDH (*n* = 3). Vehicle, Lane 1,6,11; TNF-α, Lane 2,7,12; TNF-α+ 2.5 JD, Lane 3,8,13; TNF-α+ 1.25 JD, Lane 4,9,14; TNF-α+ 0.625 JD, Lane 5,10,15. ^a^
*p* < 0.05 in comparison with vehicle.

**Figure 7 pharmaceuticals-16-00005-f007:**
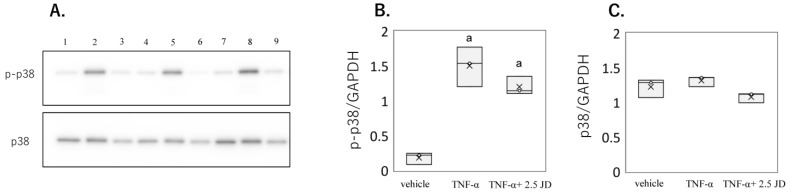
Effect of jietacin derivative on the p38 pathway in human primary synovial fibroblasts. Western blotting for p38, phosphorylated p38 (p-p38), and GAPDH (**A**). Densitometry of Western blot protein bands for p-p38 (**B**) and p38 (**C**) were normalized to the expression of GAPDH (*n* = 3). Vehicle, Lane 1,4,7; TNF-α, Lane 2,5,8; TNF-α+ 2.5 JD, Lane 3,6,9. ^a^
*p* < 0.05 compared with vehicle.

**Table 1 pharmaceuticals-16-00005-t001:** DEGs in cytokine–cytokine receptor interaction, TNF signaling pathway, NF-kB signaling pathway, and rheumatoid arthritis in the KEGG pathway.

KEGG Pathway	Gene Symbol	TNF/Vehicle		TNF + JD/TNF	
		Log2 FC	Q-Value	Log2 FC	Q-Value
C	*ACKR4*	2.12	9.25 × 10^−6^	−2.61	1.12 × 10^−6^
N	*BCL2A1*	5.13	3.58 × 10^−5^	−4.61	1.47 × 10^−91^
N/T	*BIRC3*	4.50	0	−4.79	8.37 × 10^−277^
C	*BMP4*	3.78	1.88 × 10^−38^	−3.57	3.03 × 10^−36^
C/R/T	*CCL2*	4.26	0	−4.25	4.01 × 10^−292^
C/R/T	*CCL20*	5.39	1.13 × 10^−2^	−3.57	8.66 × 10^−12^
C/R	*CCL3*	6.65	3.09 × 10^−8^	−2.10	2.45 × 10^−5^
C/R/T	*CCL5*	4.33	2.99 × 10^−111^	−3.86	5.14 × 10^−89^
C/R/T	*CSF2*	7.01	2.41 × 10^−27^	−4.92	1.42 × 10^−11^
C/T	*CXCL10*	8.52	6.82 × 10^−36^	−5.49	1.79 × 10^−7^
C	*CXCL11*	7.69	4.85 × 10^−34^	−7.12	3.62 × 10^−37^
C/R/T	*CXCL5*	2.79	1.58 × 10^−6^	−2.44	2.56 × 10^−5^
C/R/T	*CXCL8 (IL8)*	6.64	0	−2.39	2.58 × 10^−140^
C	*CXCL9*	5.60	3.51 × 10^−2^	−5.44	2.57 × 10^−2^
N	*DDX58*	3.10	1.38 × 10^−194^	−3.47	1.89 × 10^−200^
C	*EBI3*	3.11	2.40 × 10^−4^	−2.31	3.15 × 10^−3^
C	*GDF6*	−2.32	6.02 × 10^−11^	2.37	1.19 × 10^−10^
N/R/T	*ICAM1*	4.49	0	−3.73	1.07 × 10^−297^
C/R	*IL1A*	5.46	1.11 × 10^−100^	−3.68	8.14 × 10^−78^
C/N/R/T	*IL1B*	5.52	0	−4.74	0
C	*IL32*	3.65	1.25 × 10^−128^	−3.53	4.70 × 10^−101^
C	*IL34*	3.17	1.33 × 10^−10^	−3.72	1.10 × 10^−11^
C	*IL3RA*	5.86	3.61 × 10^−3^	−7.14	1.09 × 10^−4^
C/R/T	*IL6*	5.05	0	−3.47	9.57 × 10^−29^
Cyto	*INHBA*	2.09	8.81 × 10^−111^	−2.57	3.21 × 10^−128^
T	*IRF1′*	2.50	7.42 × 10^−103^	−2.41	4.09 × 10^−66^
C/T	*LIF’*	2.86	4.49 × 10^−281^	−3.41	3.37 × 10^−195^
T	*MMP9*	5.86	3.57 × 10^−3^	−4.69	1.41 × 10^−3^
N	*NFKB2*	2.74	1.33 × 10^−125^	−2.54	2.31 × 10^−91^
N/T	*NFKBIA*	3.06	1.19 × 10^−192^	−2.94	1.94 × 10^−141^
N	*PLCG2*	2.39	3.31 × 10^−2^	3.64	1.04 × 10^−28^
N	*RELB*	2.79	3.15 × 10^−2^	−2.36	7.73 × 10^−65^
R	*TLR2*	2.84	2.27 × 10^−66^	−2.69	1.49 × 10^−52^
C/N/T/R	*TNF’*	6.04	2.18 × 10^−18^	−2.22	4.41 × 10^−10^
N/T	*TNFAIP3*	4.22	0	−4.18	0
C	*TNFRSF11B*	2.38	1.44 × 10^−74^	−2.12	1.20 × 10^−52^
C	*TNFRSF9*	4.62	4.37 × 10^−53^	−4.11	2.99 × 10^−45^
C	*TNFSF10*	4.60	2.12 × 10^−4^	−5.02	2.82 × 10^−4^
C/N/R/T	*TNFSF13B*	4.33	8.79 × 10^−23^	−3.32	4.33 × 10^−17^
N/T	*TRAF1*	4.73	3.09 × 10^−237^	−4.20	1.13 × 10^−167^
N/T	*VCAM1*	5.83	1.44 × 10^−195^	−6.68	7.78 × 10^−153^

C, cytokine–cytokine receptor interaction; N, NF-κB signaling pathway; R, rheumatoid arthritis: T, TNF signaling pathway; FC, fold change.

**Table 2 pharmaceuticals-16-00005-t002:** Sequences of primers used in this study.

Primer	Sequence (5′–3′)	Product Size (bp)
*IL1B*-F	GTACCTGTCCTGCGTGTTGA	153
*IL1B*-R	GGGAACTGGGCAGACTCAAA	
*IL6*-F	GAGGAGACTTGCCTGGTGAAA	199
*IL6*-R	TGGCATTTGTGGTTGGGTCA	
*IL8*-F	ACACTGCGCCAACACAGAAA	89
*IL8*-R	CAACCCTCTGCACCCAGTTT	
*GAPDH*-F	TGTTGCCATCAATGACCCCTT	202
*GAPDH*-R	CTCCACGACGTACTCAGCG	

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
