# Peer review of "Jietacin Derivative Inhibits TNF-α-Mediated Inflammatory Cytokines Production via Suppression of the NF-κB Pathway in Synovial Cells"

_pharmaceuticals, 2022, doi:10.3390/ph16010005_

Round 1
Reviewer 1 Report
The article titled Jietacin derivative, (Z)-2-(8-oxodec-9-yn-1-yl)-1-vinyldiazene 1-2 oxide, inhibits TNF-α-mediated inflammatory cytokine and 3 chemokine expression via suppression of p65 phosphorylation 4 and importin β1 expression in synovial cell line SW982 Is accepted after consideration of the following minor comments.
.
1) Title is very long so authors should use short title
2) Abstract; abbreviations full name must mention where firstly appear (OA)
3) Page 1, line 24, z should be italic.
4) Abstract, authors should mention results.
5) Abstract , authors should mention the significant of their conclusion.
6) Page 1 line 44-45, national international what it this means.
7) it is better to merge results and discussion.
8) Page 7, line 106, i think it is already present in conclusion, so remove it.
9) discussion section concerned mainly on reported results but the discussion should discuss the obtained results.
10) conclusion should be reflect study results with interpretation.
11) It is better to improve rational of the study.
Author Response
Reviewer 1
Comments and Suggestions for Authors
The article titled Jietacin derivative, (Z)-2-(8-oxodec-9-yn-1-yl)-1-vinyldiazene 1-2 oxide, inhibits TNF-α-mediated inflammatory cytokine and 3 chemokine expression via suppression of p65 phosphorylation 4 and importin β1 expression in synovial cell line SW982 Is accepted after consideration of the following minor comments.
.
1) Title is very long so authors should use short title
Response: Thank you for reviewing our manuscript and your kind provision of comments. We have revised the title as follows:
Jietacin derivative inhibits TNF-α-mediated inflammatory cytokines production via suppression of the NF-κb pathway in synovial cells
2) Abstract; abbreviations full name must mention where firstly appear (OA)
Response: We have corrected it.(Line 21-22)
3) Page 1, line 24, z should be italic.
Response: It has been italicized.
4) Abstract, authors should mention results.
Response: We have added the results.
5) Abstract , authors should mention the significant of their conclusion.
Response: We have added the significance of the conclusions.
6) Page 1 line 44-45, national international what it this means.
Response: We have revised “national international” to “a number of”.
7) it is better to merge results and discussion.
Response: We have merged the Results and Discussion.
8) Page 7, line 106, i think it is already present in conclusion, so remove it.
Response: We have removed this sentence.
9) discussion section concerned mainly on reported results but the discussion should discuss the obtained results.
Response: We have revised the Discussion section
10) conclusion should be reflect study results with interpretation.
Response: We have revised the Conclusion section
11) It is better to improve rational of the study.
Response: We have added a sentence to improve the rational of the study.
Reviewer 2 Report
The authors found JD inhibited TNF-α-mediated inflammatory reaction via the suppression of p65 phosphorylation and decrease in importin β1 in the synovial cell line SW982.
The idea is clear, and the experimental design is also good, but the results are not enough. The following issues should be addressed.
1. The authors used the synovial cell line SW982 as the cell model, but human primary synovial cells are a better choice. And it would be much better if there were any animal experiment to verify the results.
2. The authors verified IL6 and IL8, what about IL1β ? IL1β is also a common inflammatory factor in OA.
3. How many replicates were used in RNA-seq and the following qPCR validations, which should be addressed in the manuscript.
4. Please provide access number of RNA-seq data.
5. ‘NFκB’ though the whole manuscript needs to be unified. κ is a Greek letter. In line 26, 34,40, 59, 119, 121, 155, 168, 169.
6. The symbol of the unit also needs to be revised. Like μg/mL, μL, mL.
Author Response
Reviewer 2
Comments and Suggestions for Authors
The authors found JD inhibited TNF-α-mediated inflammatory reaction via the suppression of p65 phosphorylation and decrease in importin β1 in the synovial cell line SW982.
The idea is clear, and the experimental design is also good, but the results are not enough. The following issues should be addressed.
- The authors used the synovial cell line SW982 as the cell model, but human primary synovial cells are a better choice. And it would be much better if there were any animal experiment to verify the results.
Response: We have added these data using primary synovial fibroblasts in Figure 3, 5, and 7. We agree with the issue of additional data regarding animal experiments to verify the results in vitro. We have added this point as a limitation.
- The authors verified IL6 and IL8, what about IL1β ? IL1β is also a common inflammatory factor in OA.
Response: Thank you for these valuable comments. We have added the data regarding IL-1β in Figure 2A-B.
- How many replicates were used in RNA-seq and the following qPCR validations, which should be addressed in the manuscript.
Response: Two replicate samples were taken from the vehicle, TNF-α and TNF-α +JD groups for RNA-Seq. We have added this point in the Materials and Methods section. (Line 246-247)
- Please provide access number of RNA-seq data.
Response: RNA-seq raw data were deposited in the DNA data bank of Japan (DDBJ) and have been assigned accession no. DRA015282. We have added this statement in the Data Availability Statement. (Line 340-341)
- ‘NFκB’ though the whole manuscript needs to be unified. κ is a Greek letter. In line 26, 34,40, 59, 119, 121, 155, 168, 169.
Response: ‘NFκB’ has been unified through the whole manuscript.
- The symbol of the unit also needs to be revised. Like μg/mL, μL, mL.
Response: We have carefully revised the expression of the units used through the manuscript.
Round 2
Reviewer 2 Report
1 There are two lines between NF and κB in the title.
2 Line 26, ‘to determined’ -> ‘to be determined’ ?
3 Check the grammar in line 29-33.
4 Line 198 ‘TNF-α+ 1.25 JD, Lane 5,10,15.’ ?
5 The second ‘TNF-α+ 2.5 JD’ in line 206 and 221 are unnecessary. Check all the figure legends again.
6 The number of the subtitle-Materials and Methods, Conclusion.
Author Response
1 There are two lines between NF and κB in the title.
Response: We have corrected it.
2 Line 26, ‘to determined’ -> ‘to be determined’ ?
Response: We have revised ‘to determined’ to ‘to be determined’ ?
3 Check the grammar in line 29-33.
Response: This sentence has been edited for English language by a native English speaking medical editor at DMC Corp.
4 Line 198 ‘TNF-α+ 1.25 JD, Lane 5,10,15.’ ?
Response: We have revised ‘TNF-α+ 1.25 JD, Lane 5,10,15.’ to ‘TNF-α+ 0.625 JD, Lane 5,10,15.’
5 The second ‘TNF-α+ 2.5 JD’ in line 206 and 221 are unnecessary. Check all the figure legends again.
Response: We have corrected in line 206 and 221. We have carefully checked all the figure legends again.
6 The number of the subtitle-Materials and Methods, Conclusion.
Response: We have revised the number of the subtitle-Materials and Methods, Conclusion.